# Real-Life Effectiveness of Subcutaneous Immune Therapy with Carbamylated Monomeric Allergoids on Mite, Grass, and Pellitory Respiratory Allergy: A Retrospective Study

**DOI:** 10.3390/jcm11247384

**Published:** 2022-12-12

**Authors:** Mario Di Gioacchino, Loredana Della Valle, Rocco Mangifesta, Arianna Lumaca, Francesco Cipollone, Franco Frati, Enrico Compalati, Eralda Lekli, Etleva Qirco Loloci, Francesca Santilli

**Affiliations:** 1YDA—Institute for Immunotherapy and Advanced Biological Treatments, 65100 Pescara, Italy; 2Centre of Advanced Studies and Technology (CAST), G. d’Annunzio University, 66100 Chieti, Italy; 3Pneumology Division, University Hospital, 66100 Chieti, Italy; 4Specialization School in Allergology and Clinical Immunology, G. d’Annunzio University, 66100 Chieti, Italy; 5Department of Medicine and Science of Ageing, G. d’Annunzio University, 66100 Chieti, Italy; 6Medical Department, Lofarma, 20143 Milan, Italy; 7Allergology, Salus Hospital, 1031 Tirana, Albania; 8Allergology and Clinical Immunology Service, Mother Teresa UHC, 1031 Tirana, Albania

**Keywords:** real-life study, carbamylated monomeric allergoid, SCIT, patient satisfaction, respiratory allergy

## Abstract

Background: real-life studies are encouraged to evaluate the effectiveness and safety of allergen immunotherapy (AIT). In this context, a retrospective cohort study was conducted to assess the effectiveness and safety of carbamylated monomeric allergoid subcutaneous immunotherapy (MA-SCIT), along with patient satisfaction. Methods: a total of 291 patients with rhinoconjunctivitis with or without asthma with inhalant (house dust mite, grass, and pellitory) allergies were enrolled in this study. Perceived efficacy and perceived satisfaction with MA-SCIT, symptom score by VAS, ARIA classification of rhinitis, drug consumption, number of asthma worsening episodes, and asthma symptom control were evaluated by questionnaires before, after one year, at the end of treatment, and after one or two years of MA-SCIT. Results: the overall symptom score significantly decreased over the years of MA-SCIT, irrespective of specific sensitization (*p* < 0.01). There was a substantial amelioration of rhinitis severity, with a significant reduction (*p* < 0.01) in drug use. A significant reduction was observed in the asthma symptom VAS score and asthma-worsening episodes requiring systemic steroids. None of the patients reported any severe adverse reactions. Finally, 90% of the patients reported full satisfaction with the treatment. Conclusions: the study showed that AIT with carbamylated monomeric allergoids of grass, pellitory, and mites was effective and well tolerated by patients.

## 1. Introduction

Subcutaneous immunotherapy (SCIT) was introduced in 1911 [1], but the immunological modifications responsible for allergy improvement have only been clarified in the last 30 years [2,3]. The efficacy and safety of this treatment have been widely documented in clinical trials for both subcutaneous (SCIT) and sublingual immunotherapy (SLIT), with the highest levels of EBM. Studies on allergen immunotherapy (AIT) have been discussed in many reviews and meta-analyses, [4,5,6,7,8,9,10,11] all of which support AIT as an effective, relatively safe, and well-tolerated treatment for inhalant allergic diseases. However, the relative underutilization of this effective treatment is surprising. There are many reasons for not using AIT. For example, the heterogeneity of published studies and some methodological inconsistencies may affect the applicability of the results to individual, real-life settings. Furthermore, patient selectivity in randomized clinical trials is difficult to apply to patients in real-life clinical practice, who are only partially represented by the highly selected population of RCTs [12,13,14,15]. It follows that real-life studies are required to confirm the applicability of AIT in common practice [16].

In recent years, real-life studies have been performed for AIT, demonstrating its effectiveness and safety and ensuring a better understanding of the best candidates for this therapy [16], thus leading to more widespread and appropriate AIT use. The importance of real-life studies has now been realized, and it has been recommended that real-life trials be conducted, suggesting that such studies should be introduced in the major EBM validation process for each individual product [17,18].

Over the past 5 years, real-life studies have been published on injection therapy [19,20,21,22,23,24]. They focused on adverse events, compliance, and long-term preventive effects, all showing a good profile for the allergens studied and pharmaceutical products [16]. Great importance is also attributed to retrospective studies [25].

Safety is a further aspect that limits the use of both SCIT and SLIT as in some cases, severe adverse reactions have been reported for both SCIT [26] and SLIT [27]. However, the use of carbamylated allergoids could overcome this problem. In fact, they are significantly less allergenic than native ones due to a decreased capacity to bind IgE to its specific receptor while maintaining immunogenicity and thus their therapeutic efficacy. A pharmacovigilance study [28] showed that the rate of adverse reactions to monomeric allergoid-based AIT corresponds to 0.0004% of the doses administered, both local and mild, far below the commonly reported rates for native allergen AIT products, for which anaphylactic reactions have been reported in some cases [29].

Considering these premises, this real-life retrospective cohort study was conducted to assess the effectiveness, safety, patient satisfaction, and perceived effectiveness of three consecutive years of monomeric SCIT with carbamylated monomeric allergoids (MC-SCIT) for the most important allergens (mites, grass, and pellitory) in subjects with rhinoconjunctivitis with or without asthma.

## 2. Materials and Methods

The study was conducted at the Universities of Chieti (Italy) and Tirana (Albania) and was approved by the respective ethics committees.

### 2.1. Inclusion/Exclusion Criteria

All patients with rhinoconjunctivitis with or without asthma mono-sensitized to house dust mites, grass, or pellitory who had received MC-SCIT (Lofarma, Milan, Italy) for three consecutive years and had withdrawn from the treatment for 1–2 years were eligible for the study. The exclusion criteria were polysensitization to other allergens and previous AIT.

Allergy was diagnosed in all patients using the skin prick test (Lofarma, Milan, Italy) and specific IgE. All patients with asthma underwent pulmonary function tests before treatment and almost every 12 months during the AIT and follow-up period.

### 2.2. Treatment

SCIT had been prescribed according to international guidelines [30,31]. The dosage and intervals of administration were established on the technical data sheet for the product (Appendix A). In particular, after a building phase with an increasing dose (0.1 to 0.5 mL) administered every week, a maintenance dose of 0.5 mL was administered every month for 3 years. Every maintenance shot of mites (0.5 mL) contained 2 μg of group 1 major allergens (Der p 1 and Der f 1), those of grass contained 2.5 μg of group 5 major allergens (Phl p5, Hol l5, and Poa p5), and those of pellitory contained 0.75 μg of group 1 major allergens (Par j 1 and Par o 1).

Treatments were performed at the allergy clinics in 48% of the patients, at the family doctors in the remaining cases. In either case, a control visit at the allergy unit was performed every year or as needed on request.

All anti-allergic treatments, such as inhaled and oral steroids, antihistamines, and beta2 agonists, were prescribed by the allergy specialist at various control visits. All patients were given written instructions to treat their exacerbations and to record the drugs used and their dosage.

### 2.3. Questionnaires

Specific digital questionnaires for asthma and rhinoconjunctivitis were prepared (Appendix A) and administered to all patients with questions related to four time points: the year before the start of MC-SCIT (T0), the end of the first year of MC-SCIT (T1), the end of the three years of MC-SCIT (T2), and one–two years after discontinuation of MC-SCIT (T3).

In particular, patients were asked questions about (a) perceived efficacy of the treatment measured through a digital VAS for symptoms with a score from 0 to 10, where 0 represents “no symptoms” and 10 represents “severe and extremely annoying symptoms”; (b) local and systemic antihistamine and steroid treatment (oculorhinitis patients) evaluated by a digital VAS where 0 represents “never used” and 10 represents “frequently used;”; (c) ARIA classification of rhinitis based on the criteria of duration and severity/impact of symptoms on the quality of life [32,33] (data relating to T0 and T3); (d) assessment of asthma control by means of a standardized questionnaire concerning the frequency of daytime and nocturnal asthma symptoms, use of reliever medications, and limitation of any activity due to asthma; (e) number of severe asthma attacks requiring systemic steroid treatment or hospitalization; and (f) perceived impact of rhinitis and asthma using the rhinitis asthma patient perspective (RAPP) test [34]. Moreover, perceived satisfaction with the treatment expressed through one of the following options: “very dissatisfied,” “dissatisfied,” “satisfied,” and “very satisfied” was evaluated at the end of the study.

The data were obtained online (giving telephonic assistance through a professional, where necessary) in view of the concurrent COVID-19 pandemic using software that guaranteed the anonymity and integrity of the data acquired and compliance with privacy legislation. All patients signed an informed consent form to participate in the study and were informed of the anonymity of the data acquired.

### 2.4. Statistical Analysis

All quantitative variables are summarized as mean and standard deviation, and the qualitative variables are summarized as frequency and percentage. Statistical analyses were performed using non-parametric tests. The chi-square test for unpaired data was used to compare the qualitative variables between the Albanian and Italian patients, while the Mann–Whitney *U* test for paired data was applied to evaluate the differences in various parameters between the same groups. The Wilcoxon test was applied for comparison before and after treatment. The Friedman test was used to evaluate differences in VAS scores within the same group. McNemar’s test was applied to the paired nominal data. Cochran’s *Q* test was applied where the response variable only had two possible outcomes. The statistical significance of the differences was evaluated at an alpha level of 0.05. Statistical analysis was performed using the SPSS software (version 11.0; SPPS Inc., Chicago, IL, USA).

## 3. Results

A total of 291 patients (169 Albanian and 122 Italian) were enrolled in this study.

### 3.1. Demographics

No demographic differences were found between the Albanian and Italian patients (Chi-square test: *p* = 0.215; Mann–Whitney U-Test: *p* = 0.811). Rhinoconjunctivitis and sensitization to dust mites were the most frequent in both groups but were more prevalent in the Albanian group (Table 1).

### 3.2. Baseline Characteristics

Before MA-SCIT, according to the ARIA classification, 109 patients had severe/persistent rhinitis (22 grass-sensitized, 1 pellitory-sensitized, and 86 mite-sensitized), 128 had severe/intermittent rhinitis (40 grass-sensitized, 19 pellitory-sensitized, and 71 mite-sensitized), 13 had mild/persistent rhinitis (6 grass-sensitized, 2 pellitory-sensitized, and 5 mite-sensitized), and 12 had mild/intermittent rhinitis (9 grass-sensitized, 0 pellitory-sensitized, and 4 mite-sensitized).

Furthermore, 60 of 67 asthmatic patients had uncontrolled asthma before treatment, since they reported frequent symptoms, activity limitation, sleep disorders, and need for asthma relievers, with an average of 3.5 exacerbations per year.

### 3.3. MC-SCIT Effectiveness

#### 3.3.1. Overall Results

The perception of MC-SCIT effectiveness of the patients, as assessed by symptom reduction over the course of treatment, was excellent. The symptom score evaluated by VAS decreased over the years (Figure 1A) with a high statistical significance (Friedman test: *p* < 0.001), with a reduction in symptom score evident and significant already after the first year of MC-SCIT (Wilcoxon test: *p* < 0.01). Similar results were obtained by dividing the patients according to clinical manifestation (Figure 1B) or specific sensitization (Table 2).

#### 3.3.2. Patients with Oculorhinitis

The decrease in symptom scores in patients with oculorhinitis confirmed a reduction in the severity of rhinitis, as evaluated by the parameters of the ARIA document. There was a significant reduction in the persistence and severity of symptoms, sleep disturbance, and symptom interference with daily activities after treatment compared with before treatment (Table 3). It follows that there was a decrease in the number of patients with severe/persistent (from 109 to 10) and severe/intermittent (from 128 to 94) rhinitis, in parallel with an increase in those with mild rhinitis (from 28 to 161), the majority was intermittent (141).

Similar results were obtained by dividing patients according to specific sensitization. A reduction in severe and persistent rhinitis was paralleled by an increase in mild and intermittent rhinitis for sensitization to all three allergens. In particular, the number of patients with severe rhinitis decreased from 62 before to 17 after treatment in grass-sensitized, from 20 to 2 in pellitory-sensitized, and from 157 to 85 in mite-sensitized patients, with an increase in mild rhinitis from 15 to 60 after treatment in grass-sensitized, from 2 to 20 in pellitory-sensitized, and from 9 to 81 in mite-sensitized patients.

The consumption of local and systemic antihistamines and steroids, evaluated by VAS, over the course of MC-SCIT was in line with the reported symptom decrease, with a significant reduction during the first year of MC-SCIT (Wilcoxon: *p* < 0.01) and throughout the treatment period (Friedman test: *p* < 0.001) (Figure 2A). A significant reduction in drug consumption was also detected in patients with allergic sensitization (Friedman test: *p* < 0.01 in patients allergic to pellitory; *p* < 0.001 in patients allergic to grass and mites). Significantly lower drug use was observed even after the first year of MC-SCIT in all three groups of patients (Wilcoxon test: *p* < 0.05) (Figure 2B).

#### 3.3.3. Patients with Asthma

The improvement of symptoms was also evident in patients with asthma, whether associated or not with oculorhinitis, with a significant improvement in VAS over the year of MC-SCIT (Friedman test: *p* < 0.001) (Figure 1B). VAS reduction was also observed in patients based on their allergic sensitization; the reduction of VAS observed over the years of MC-SCIT reached a significance of *p* < 0.01 (Friedman test) in patients allergic to mites and pellitory and *p* = 0.05 in patients allergic to grass (Table 2). Patients with mite and pellitory allergies also showed a significant VAS reduction after the first year of MC-SCIT compared with that before treatment (*p* < 0.05), whereas the change in VAS did not reach statistical significance in patients with grass allergies.

Asthma improvement was confirmed by the significant reduction in asthma exacerbations per year requiring treatment with systemic corticosteroids, considering all patients together or by dividing them by allergic sensitization (Table 4).

All patients were requested to answer the questions on the asthma control test referred to at various stages of the treatment, in particular, frequency of symptoms, activity limitation, sleep disorder, and need for relievers due to asthma. In all cases, or by dividing patients by allergic sensitization, there was a reduction in all parameters with a high statistical significance (Cochran’s Q test: *p* < 0.001) (Figure 3).

#### 3.3.4. Results 1–2 Years after MC-SLIT

Patients, considered together or divided by disease or allergic sensitization, maintained the improvements found at the end of therapy 1–2 years after the end of MC-SLIT (all Figures and Tables). Moreover, patients with asthma and oculorhinitis were asked to answer the RAPP questionnaire, all but two showed values lower than 15, demonstrating good disease control (Table 5).

The vast majority (90%) of the patients, regardless of the disease or specific sensitization, declared their satisfaction with the treatment received; five patients were unhappy, one was dissatisfied, and four were disappointed (Table 6).

### 3.4. Adherence to Treatment

Sixty percent of the patients completed the course of MA-SCIT with a maximum delay of five days in carrying out various maintenance doses. Twenty missed up to three injections during the course of therapy, eleven missed up to five injections, and nine missed up to seven injections

During the 3 years of treatment, the maximal dose received by patients with mite allergies was 74.4 μg of major allergens (Der p 1 and Der f 1), patients with grass allergies received 93 μg (Phl p5, Hol l5, and Poa p5), and patients with pellitory allergies received 27.9 μg (Par j 1 and Par o 1).

### 3.5. Safety

No patient reported severe local and systemic adverse reactions, and only 12 patients reported a local reaction at the site of injection, not requiring treatment.

## 4. Discussion

This study showed that carbamylated monomeric allergoid-based SCIT is a safe and effective treatment for grass, pellitory, and mite respiratory allergies.

The retrospective assessments in this study were performed using VAS for various items. The VAS is well-validated for the measurement of rhinitis and asthma symptoms and correlates well with the severity assessed by various questionnaires (ARIA, rTNSS, RQLQ, and ACT). It has been used in several treatment studies, demonstrating that VAS is highly effective in assessing disease control [35,36,37].

The concordance of the various results obtained in the present study confirms the accuracy of the results. The significant improvement in VAS for oculorhinitis symptoms correlates with the decrease in the severity of rhinitis over the year of MC-SCIT, evaluated by the ARIA questionnaire, and results corroborated by the significant reduction in the use of local and systemic antihistamines and steroids. The improvement in these parameters was progressive during the treatment, with early amelioration after the first year of MC-SCIT. These results were consistent when considering all recruited patients as well as selecting them according to the presence or absence of asthma or allergic sensitization, namely grass, pellitory, and mites.

Similar results were obtained in asthmatic patients (all patients together or divided by allergic sensitization), where the improvement of VAS for symptoms correlated with reduction of sleep disorders, activity limitation, frequency of symptoms, need for a reliever, and severe exacerbations over the year of MC-SCIT, in all cases with a high statistical significance.

The good clinical condition reached at the end of MC-SCIT was maintained 1–2 years after the end of treatment, as demonstrated by the almost perfect equivalence between values found at T2 and those found at T3 (considering all patients together or divided by clinical manifestation or allergic sensitization), confirming that the effect of MC-SLIT persisted after the end of treatment [38]. Health status was confirmed in patients with asthma and oculorhinitis by the RAPP questionnaire values, which were lower than 15 in all patients except two. Values lower than 15 demonstrated good asthma and rhinitis control [34].

Considering both the perceived health status of the patients (VAS for symptoms, level of sleep, and interference with daily activities) and the final questionnaire on their level of satisfaction with the treatment received, it may be concluded that the expectations of the patients are adequately fulfilled for all three sensitizing allergens. Patient satisfaction is a useful measure of the effectiveness of medical treatment. It affects health-related decisions and treatment-related behaviors of the patients, which in turn have a substantial effect on the success of treatment outcomes in many fields of medicine [39,40]. Patient satisfaction with AIT was recorded and was a key factor in its long-term efficacy assessment and adherence to therapy [41,42,43,44].

Data from large-scale real-life trials on subcutaneous AIT [19,20,21,22,23,45,46,47,48,49] demonstrate that SCIT in a real-world setting is effective and achieves significantly higher adherence rates than sublingual administration [50]. Adherence to AIT can be increased by minimizing its adverse reactions. The patients participating in the present study did not experience any severe adverse reactions and the few reported were local, mild, and did not require treatment. The safety profile of the allergoids used in this study is based on chemical modification by monomeric carbamylation, which causes allergen lysine substitution with preserved size and structural conformation of the native allergen but with a low capacity to link IgE to its specific receptor [51], while maintaining immunogenicity [52,53,54,55]. There are many reports showing that both SLIT and SCIT with carbamylated monomeric allergens are safe as well as effective [28,56,57,58,59,60].

Limitations of the study: although the number of enrolled patients was largely sufficient to evaluate the global effectiveness of MA-SCIT, selecting patients by specific sensitization, the number of pellitory allergic subjects should be increased to confirm the results obtained, even if they are already statistically significant. Equally, the number of asthmatic pollen-sensitized patients evaluated was small, in any case the evaluation of the entire group of asthmatics supports the conclusion reached.

Moreover, it was impossible to examine drug consumption by checking drug prescription records, which would render the data more objective. However, the accordance of all the results obtained in this study minimizes this shortcoming.

Conclusion and key messages:

This study shows that carbamylated monomeric allergoid-based SCIT is effective in treating respiratory allergies to mites, grass, and pellitory. Moreover, this therapy does not cause severe local or systemic adverse reactions.

## Figures and Tables

**Figure 1 jcm-11-07384-f001:**
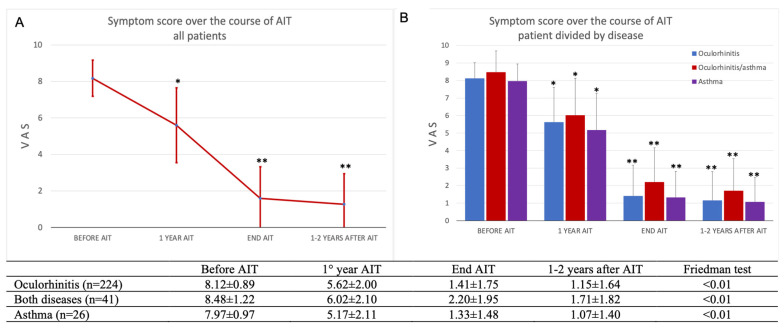
Symptom score evaluated by VAS over the years of AIT. A significant decrease of VAS was observed in all patients (291) (**A**), and selecting patients with oculorhinitis, asthma, or both (**B**). Values are expressed as (µ ± SD). * *p* < 0.01 Wilcoxon test comparing VAS changes at 1 year AIT respect before. ** *p* < 0.01 Friedman test detecting differences across the course of AIT.

**Figure 2 jcm-11-07384-f002:**
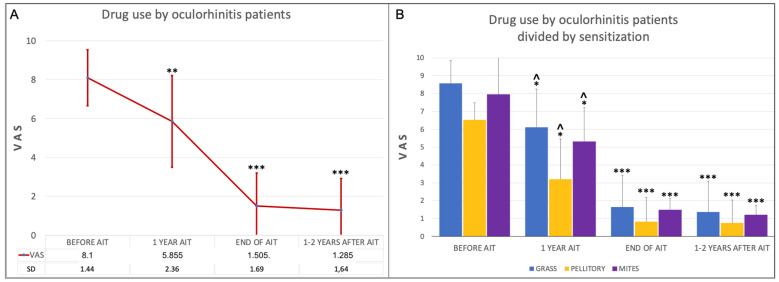
(**A**). VAS for the use of oral and systemic antihistamines and corticosteroids by all patients with oculorhinitis (265) over the course of AIT. A significant decrease was observed comparing the various phases of AIT with respect to before the treatment. ^ *p* < 0.01 Wilcoxon test comparing VAS changes at 1 year AIT respect before. * *p* < 0.001 Friedman test detecting differences across the course of AIT. (**B**). Drug consumption (systemic and local antihistamines and steroids) by oculorhinitis patients divided by allergic sensitization over the course of AIT. ^ *p* < 0.05 Wilcoxon test comparing VAS changes at 1 year AIT with respect to before treatment. * *p* < 0.01; ***p* < 0.05; *** *p* < 0.001 Friedman test detecting differences across the course of AIT.

**Figure 3 jcm-11-07384-f003:**
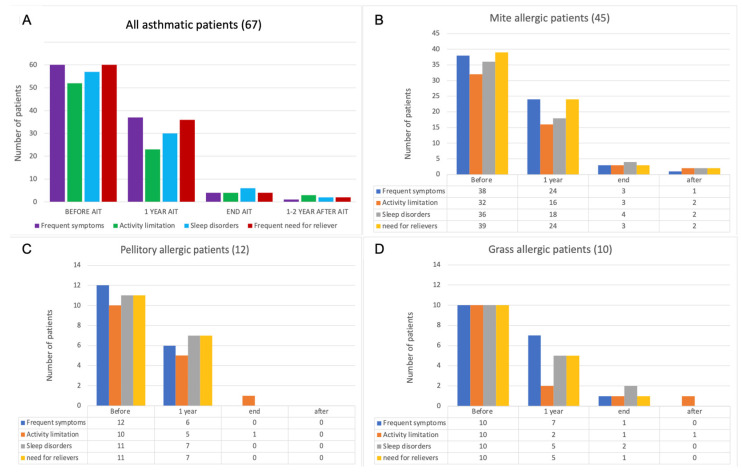
Number of patients with specific symptoms indicative of asthma control. (**A**) all asthmatics, (**B**) mite allergic patients, (**C**) pellitory allergic patients, and (**D**) grass allergic patients. Statistical significance of *p* < 0.001 in all cases (Cochran’s Q test).

**Table 1 jcm-11-07384-t001:** Demographic and allergological characteristics of enrolled patients.

	169 Albanian Pts	122 Italian Pts	Total: 291
Sex *, men/women	114/55	76/46	190/101
Age **, years *± SD*	26.6 ± 13.0	26.9 ± 14.4	26.7 ± 13.6
Disease ***, no. (%)			
Asthma	7 (4.1)	19 (15.5)	26 (8.9)
Rhinoconjunctivitis	142 (84.1)	82 (67.2)	224 (77.0)
Asthma and Rhinoconjunctivitis	20 (11.8)	21 (17.2)	41 (14.1)
Type of Allergen ***, no. (%)			
Dust mite	105 (62.1)	77 (63.1)	182 (62.5)
Grass	62 (36.7)	17 (13.9)	79 (27.1)
Pellitory	2 (1.2)	28 (23.0)	30 (10.3)

* *p* = 0.215, Chi-square test; ** *p* = 0.811, Mann-Whitney U-Test; *** *p* < 0.01, Chi-square test.

**Table 2 jcm-11-07384-t002:** VAS changes in AIT-treated patients divided by allergic sensitization.

	Before AIT	1° Year AIT	End AIT	1–2 Years after AIT	*p* Value*
Grass allergic patients					
All oculorhinitis (n = 77)	8.46 ± 0.91	6.65 ± 1.90 ^	2.10 ± 1.99	1.72 ± 1.90	<0.01
All asthmatics (n = 10)	8.60 ± 0.40	7.00 ± 1.63 ns	3.01 ± 1.18	2.90 ± 1.11	0.05
Pellitory allergic patients					
All oculorhinitis (n = 22)	8.57 ± 0.75	5.32 ± 1.20 ^	1.01 ± 1.01	1.01 ± 1.08	<0.01
All asthmatics (n = 12)	8.54 ± 1.1	5.45 ± 1.44 ^	0.90 ± 0.98	0.69 ± 0.60	<0.01
Mite allergic patients					
All oculorhinitis (n = 166)	8.09 ± 0.96	5.27 ± 1.95 ^	1.70 ± 1.90	0.98 ± 1.68	<0.01
All asthmatics (n = 45)	8.13 ± 1.05	5.54 ± 2.20 ^	1.88 ± 1.78	1.42 ± 1.62	<0.01

Values expressed as µ ± SD. * Friedman test over the course of treatment; ^ *p* < 0.05 Wilcoxon test: 1 year AIT versus before; ns = not significant.

**Table 3 jcm-11-07384-t003:** Aria classification of rhinitis over the years of AIT (n. 265 pts).

	Before Therapy			After Therapy			*p* *
	Yes: n (%)	No: n (%)	Never	Yes: n (%)	No: n (%)	Never	
All patients (265)							
Symptoms > 4 d/w–4 week	122 (46)	143 (54)	0	30 (11.3)	28 (10.6)	207 (78.1)	<0.001
Troublesome symptoms	237 (89.4)	28 (10.6)		7 (2.6)	258 (97.4)		<0.001
Impairment of daily activity	173 (65.3)	92 (34.7)		61 (23)	204 (77)		<0.001
Sleep disturbance	219 (82.6)	46 (17.4)		21 (7.9)	244 (92.1)		<0.01
Grass sensitized (77)							
Symptoms > 4 d/w–4 week	28 (36.4)	49 (63.6)	0	13 (16.9)	13 (16.9)	51 (66.2)	<0.001
Troublesome symptoms	60 (77.9)	17 (22.1)		3 (3.89)	74 (96.1)		<0.001
Impairment of daily activity	54 (70.1)	23 (29.9)		17 (22.1)	60 (77.9)		<0.001
Sleep disturbance	62 (80.5)	15 (19.5)		15 (19.5)	62 (80.5)		<0.01
Pellitory sensitized (n. 22)							
Symptoms > 4 d/w–4 week	3 (13.6)	19 (86.4)	0	1 (4.5)	0	21 (95.5)	<0.001
Troublesome symptoms	20 (90.9)	2 (9.1)		2 (9.1)	20 (90.9)		<0.001
Impairment of daily activity	16 (72.7)	6 (27.3)		2 (9.1)	20 (90.9)		0.01
Sleep disturbance	13 (59.1)	9 (40.9)		2 (9.1)	20 (90.9)		<0.05
Mite sensitized (166)							
Symptoms > 4 d/w–4 week	91 (54.8)	75 (45.2)	0	16 (9.6)	15 (9.0)	135 (81.3)	<0.001
Troublesome symptoms	157 (94.6)	9 (5.4)		2 (1.2)	164 (98.8)		<0.01
Impairment of daily activity	103 (62.0)	63 (37.97)		42 (25.3)	124 (74.7)		<0.01
Sleep disturbance	144 (86.7)	22 (13.3)		4 (2.4)	162 (97.6)		<0.001

* McNemar Test, before therapy vs. after therapy.

**Table 4 jcm-11-07384-t004:** Number of asthma exacerbations requiring systemic steroids over the course of treatment.

	Before AIT Me (Min–Max)	1° year AIT Me (Min–Max)	End AIT Me (Min–Max)	1–2 y after AIT Me (Min–Max)	Friedman Test
All asthmatics	3.5 (2–8)	1.5 (0–7)	0.5 (0–2)	0.5 (0–2)	<0.01
Grass allergic (n = 10)	3.5 (2–5)	1.5 (0–4)	0.5 (0–2)	0.5 (0–2)	0.05
Pellitory allergic (n = 12)	3.5 (0–8)	1.5 (0–3)	0.5 (0–2)	0.0 (0–1)	<0.01
Mite allergic (n = 67)	3.0 (0–8)	0.5 (0–7)	0.0 (0–2)	0.0 (0–2)	<0.01

**Table 5 jcm-11-07384-t005:** RAPP evaluation after treatment in patients with oculorhinitis and asthma.

	Mean	SD	Median	Min–Max
Mites (n. 29)				
RAPP value	11.7	4.6	10	8–24
Grass (n. 8)				
RAPP value	11.7	2.2	11.5	9–15
Pellitory (n. 4)				
RAPP value	9.5	4.0	9.5	8–11
All patients (n. 41)				
RAPP value	11.49	4.02	10	8–24

**Table 6 jcm-11-07384-t006:** Patient perceived health status and satisfaction at the end of MA-SCIT.

Satisfaction	Disease		
	Asthma	Rhinoconjunctivitis	Both
Completely satisfied	26 (100%)	218 (97.4)	37 (90.2)
Unhappy	--	3 (1.3)	2 (4.9)
Unsatisfied	--	1 (0.4)	--
Disappointed	--	2 (0.9)	2 (4.9)
Total	26	224	41

## Data Availability

The data used in this research are not publicly available due to concerns about confidentiality; however, we summarize descriptive information about our participants in the manuscript. The data that support the findings of this study are available upon reasonable request.

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
