# Peer review of "Real-Life Effectiveness of Subcutaneous Immune Therapy with Carbamylated Monomeric Allergoids on Mite, Grass, and Pellitory Respiratory Allergy: A Retrospective Study"

_jcm, 2022, doi:10.3390/jcm11247384_

Round 1
Reviewer 1 Report
I thank the authors of the article, and have some comments on this article.
1. There are so many tables. It would be better to reorganize the tables, and these should be also corrected more concisely in the manuscripts.
2. It is confusing to understand the article because there were so many contents.
3. The efficacy of AIT such as drug use (antihistamine, systemic steroid, reliver for asthma), asthma exacerbations (hospitalization) was evaluated using questionnaire in this study. However, it will be more objective and accurate to check the records of drug prescription and hospitalization.
4. Check the format of the manuscript: Figure, table such as fonts, case letters (upper/lower)…
These corrections should be made and then the article should be reviewed.
Author Response
The authors thank the reviewer for his suggestions.
Below are the responses to the comments
- There are so many tables. It would be better to reorganize the tables, and these should be also corrected more conciselyin the manuscripts.
- It is confusing to understand the article because there were so many contents.
Response 1 and 2: The results are modified accordingly reorganizing the table and figures to make easier the riding and understanding of the manuscript. In the present form there are 3 figures instead of 6 and 6 tables respect 10 in the original manuscript.
- The efficacy of AIT such as drug use (antihistamine, systemic steroid, reliver for asthma), asthma exacerbations (hospitalization) was evaluated using questionnaire in this study. However, it will be more objective and accurate to check the records of drug prescription and hospitalization.
Response 3: Unfortunately, it was impossible to check the records of drug prescription and hospitalization. So we reported and discussed only the data coming from patients, that in any case are well matched with the other results. We reported this issue among the limitation of the study.
- Check the format of the manuscript: Figure, table such as fonts, case letters (upper/lower)…
Response 4: The format has been carefully checked
Reviewer 2 Report
The article is not properly structured; the materials and methods are imprecise; there are no clear inclusion criteria (no referral to diagnostic of specific sensitisations or pulmonary function tests); no exclusion criteria is mentioned.
Regarding the allergen immunotherapy treatment, there is no reference to allergen shots, highest concentration of the shot, maintenance dose.
The results are not structured (demographics, baseline characteristics, treatment efficacy, safety, compliance). There is only one short statement on safety concerns (132-133); no study limitation is mentioned.
Overall, although the data collected were analyzed and statistical processed, the article needs major changes.
Author Response
The authors thank the reviewer for his helpful suggestions
- The article is not properly structured; the materials and methods are imprecise; there are no clear inclusion criteria (no referral to diagnostic of specific sensitisations or pulmonary function tests); no exclusion criteria is mentioned.
Response 1: The manuscript has been structured, we added a section on inclusion and exclusion criteria and referred to diagnostics
- Regarding the allergen immunotherapy treatment, there is no reference to allergen shots, highest concentration of the shot, maintenance dose.
Response 2: Allergen immunotherapy schedule and dosage (in the previous work reported only in the supplementary file 1) have been reported in more details in the manuscript
- The results are not structured (demographics, baseline characteristics, treatment efficacy, safety, compliance). There is only one short statement on safety concerns (132-133); no study limitation is mentioned.
Response 3: The section results has been structured in demographics, baseline characteristics, treatment efficacy, safety and compliance. Moreover, the limitations of the study have been reported in the discussion.